# A stakeholder group assessment of interactions between child health and the sustainable development goals in Cambodia

Daniel Helldén [1]✉, Thy Chea[2], Serey Sok[3], Linn Järnberg[4], Helena Nordenstedt[1], Göran Tomson[5,6], Måns Nilsson[4,7] & Tobias Alfvén[1,8]

## Abstract

**Background** With the implementation of the Sustainable Development Goals, a systematic assessment of how the goals influence child health and vice versa has been lacking. We aimed to contribute to such an assessment by investigating the interactions between child health and the Sustainable Development Goals in Cambodia.

**Methods** Based on the SDG Synergies approach, 272 interactions between 16 Cambodian Sustainable Development Goals and child health were evaluated by an interdisciplinary Cambodian stakeholder group. From this a cross-impact matrix was derived and network analysis applied to determine first and second-order effects of the interactions with a focus on child health.

**Results** We show that with the exception of Cambodian Sustainable Development Goal 15 (life on land) the interactions are perceived to be synergistic between the child health and the Cambodian Sustainable Development Goals, and progress on Cambodian Sustainable Development Goal 16 (peace, justice and strong institutions) could have the largest potential to contribute to the achievement of the Cambodian Sustainable Development Goals, both when it comes to first and second-order interactions.

**Conclusions** In this stakeholder assessment, our findings provide novel insights on how complex relationships play out at the country level and highlight important synergies and trade-offs, vital for accelerating the work toward the betterment of child health and achieving the Sustainable Development Goals.

## Plain language summary

The Sustainable Development Goals (SDGs) are a set of 17 global goals set by the United Nations to guide the world toward development that meets the needs of the present without compromising the ability of future generations to meet their own needs. The efforts to achieve the different SDGs are interconnected. To better understand in what way, a group with different expertize and perspectives was assembled in Cambodia to score the linkages between the SDGs and child health. This identified that most goals promote better child health and that advancements in child health also help achieve the SDGs in Cambodia. Our study provides useful knowledge and a practical approach for policy makers trying to accelerate the work toward better child health in Cambodia.

[1] Department of Global Public Health, Karolinska Institutet, Tomtebodavägen 18A, SE-171 77 Stockholm, Sweden. [2] Malaria Consortium, Phnom Penh, Cambodia. [3] Research Office, Royal University of Phnom Penh, Phnom Penh, Cambodia. [4] Stockholm Environment Institute, Stockholm, Sweden. [5] Presidents Office, Karolinska Institutet, Stockholm, Sweden. [6] Swedish Institute for Global Health Transformation (SIGHT), Royal Swedish Academy of Sciences, Stockholm, Sweden. [7] Department of Sustainable Development, Environmental Science and Engineering, KTH Royal Institute of Technology, Stockholm, Sweden. [8] Sachs' Children and Youth Hospital, Stockholm, Sweden. ✉email: daniel.hellden@ki.se

The world has experienced an impressive decline in global child mortality over the last decades, however still 5.2 million children die before they reach their 5th birthday[1]. The 17 Sustainable Development Goals (SDGs) represent the global community's most comprehensive and people-centered set of universal targets to date that have been endorsed by governments[2]. The health and well-being of children stand to benefit, stagnate, or regress depending on progress in other sectors of society toward the attainment of the SDGs. It is nearly impossible to untangle the health of children from their social, natural and economic environments[3]. For example, it has been demonstrated that approximately half of the reduction in under-five mortality between 1990 and 2010 can be attributed to investments outside of the health sector[4].

Without losing sight of the unfinished progress on reducing global child mortality, the global strategy for women´s, children´s and adolescents' health implore researchers and decision makers to aspire beyond a world in which all children not only *survive* but *thrive* in order to realize their potential to *transform* communities[5]. Social, economic, political, environmental, and cultural determinants have important effects on child health[6–8] while the survival, health and well-being of children are crucial to reach multiple sustainable development outcomes[9,10].

The SDGs are presented in the 2030 Agenda as integrated, indivisible and interdependent and can be seen as a large system of goals that interact and affect each other directly and indirectly. However, the 2030 Agenda does not attempt to identify or characterize the interactions. A field of SDG interactions studies has emerged where a range of mostly quantitative methods have been applied to try to distinguish these interactions and subsequent network effects[11]. One such method, the SDG Synergies, a semi-quantitative participatory approach originally developed by the International Science Council and the Stockholm Environment Institute[12–14] can be used for untangling the direct and indirect effects of interactions between the SDGs. Through the scoring of relevant interactions by a multidisciplinary stakeholder group, the method allows for context-specific analysis of interactions since these vary in position and nature depending on the context within which the interaction occurs[15]. Furthermore, the framework can serve as a basis for more complex analysis and visualization of the interactions through network analysis[14,16]. The approach has previously been applied in a variety of policy contexts, ranging from global policy issues such as climate change to interactions within a specific country[12,14,16–19]. Using this approach, Blomstedt et al.[20] showed that several SDGs, including SDG 1 (no poverty), 2 (zero hunger), 4 (quality education), 5 (gender equality), 8 (decent work and economic growth) and 17 (partnership for the goals) have strong and reciprocal links with child health. Their theoretical analysis also suggested that multisectoral collaboration on some targets are essential for sustainable progress on child health, while it found few negative interactions indicating the limited number of trade-offs with health. The method quantifies expert opinions through the scoring of the interactions, and although the subjectivity of the SDG Synergies approach can be in contrast to the classical paradigm of rational and data driven decision making[21,22], real world prioritization processes are influenced by many different factors and biases[16,23,24]. To some extent, the SDG Synergies approach integrates real world human behavior into prioritization and decision making models which is necessary for understanding complex context dependent systems[24,25], forming a bridge across sectors and promoting evidence informed policy, particularly given the absence of quality quantitative data to assess the SDGs[26].

Cambodia was among the few low- and middle-income countries that achieved the Millennium Development Goal 4 and reduced the under-five mortality from 116 to 27 deaths per 1000 live births between 1990 and 2019[1,27]. However, an estimated 12,000 children still die from preventable causes every year and mortality rates among low income, less educated and more rural populations have not declined as much[1,28]. Investments outside of the health sector in education, nutrition, water and sanitation, and poverty reduction measures together with multisectoral planning and collaborative initiatives between non-health sectors have been key to the betterment of child health in Cambodia[29,30]. However, multidimensional poverty and non-monetary deprivation such as overcrowded housing, suboptimal water and sanitation facilities and lack of school attendance are still prevalent with almost half of all children under 18 years of age experienced three or more deprivations in 2018[31]. The development and adoption of the Cambodia Sustainable Development Goals (CSDGs) with its 18 goals and 88 targets offers a comprehensive framework for sustainable development localized to the country context and holds the promise of delivering for children in Cambodia[32]. The country has improved the health and well-being of children, however the role of different sectors in this achievement has not been systematically assessed. Furthermore, the interactions between the SDGs and child health have not been examined at a country level before. The aim of this study was therefore to contribute to such an assessment by determining the strength, position and nature of interactions between the SDGs and child health in Cambodia. We show that with the exception of CSDG 15 (life on land) the interactions are perceived to be synergistic between the child health and the CSDG, and progress on CSDG 16 (peace, justice and strong institutions) could have the largest potential to contribute to the achievement of the CSDGs, both when it comes to first and second-order interactions.

## Methods

The semi-quantitative SDG Synergies approach[14], applied to the Cambodia national-level context and with a primary focus on child health was utilized in this study. In brief, the SDG Synergies approach follows three overarching stages that enable the investigation of the strength, position and nature of interactions in a network, as outlined below. Further, we provide some additional analysis to ground the results in the country context.

**Identification of goals**. Between the 169 targets of the SDGs there are almost 300,000 possible pairwise interactions, hence the first step is to limit the scope of the analysis and select the goals or targets of interest. Through matching SDG priorities with national developmental goals, ministry consultations and investigations into possible data sources, the Royal Government of Cambodia has put forward the CSDGs as 18 nationalized goals and a localized set of 88 targets from the 2030 Agenda. On a goal level, the CSDGs include one additional goal (number 18) on the ending of the negative impact of Mine/Explosive remnants of war (ERW) and promote victim assistance, while the targets for each goal are fewer but designed so that data indicators can be obtained to measure the progress toward the targets[32]. Guided by CSDGs[32], the analysis done by Blomstedt et al.[20] and the relevant SDG targets identified by UNICEF[33] as well as in-depth discussions within the research team and with local partners to ensure relevancy to the Cambodian context, it was considered most adequate to include all CSDGs with the exception of CSDG 17 (partnerships for the goals) since the goal was deemed too broad for meaningful assessment. It was further decided to limit CSDG 3 (good health and well-being) to only representing child health, which we defined as a state of complete physical, mental and social well-being and not merely the absence of disease or

**Table 1 List of included Cambodia sustainable development goals and their definitions.**

| CSDG Goal | Definition |
|---|---|
| 1 | End poverty in all its forms everywhere |
| 2 | End hunger, achieve food security and improved nutrition and promote sustainable agriculture. |
| 3 | Child health |
|  | In line with the WHO definition of health and the United Nations Convention on the Rights of the Child, child health is defined as a *state of complete physical, mental and social well-being and not merely the absence of disease or infirmity among human beings below 18 years.* |
| 4 | Ensure inclusive and equitable quality education and promote lifelong learning opportunities for all. |
| 5 | Achieve gender equality and empower all women and girls. |
| 6 | Ensure availability and sustainable management of water and sanitation for all. |
| 7 | Ensure access to affordable, reliable, sustainable and modern energy for all. |
| 8 | Promote sustained, inclusive and sustainable economic growth, full and productive employment and decent work for all. |
| 9 | Build resilient infrastructure, promote inclusive and sustainable industrialization and foster innovation. |
| 10 | Reduce inequality within and among countries. |
| 11 | Make cities and human settlements inclusive, safe, resilient and sustainable. |
| 12 | Ensure sustainable consumption and production patterns. |
| 13 | Take urgent action to combat climate change and its impacts |
| 14 | Conserve and sustainably use the oceans, seas and marine resources for sustainable development. |
| 15 | Protect, restore and promote sustainable use of terrestrial ecosystems, sustainably manage forests, combat desertification, and halt and reverse land degradation and halt biodiversity loss. |
| 16 | Promote peaceful and inclusive societies for sustainable development, provide access to justice for all and build effective, accountable and inclusive institutions at all levels. |
| 18 | End the negative impact of Mine/Explosive remnants of war (ERW) and promote victim assistance. |

infirmity among human beings below 18 years. The list of CSDGs and their definitions are detailed in Table 1. The selection led to a total of 17 goals, translating into 272 interactions.

**Assessing the interactions**. Over a 2-day workshop on the 24–25th of August 2020, taking place in Phnom Penh, 29 participants representing a range of governmental and non-governmental stakeholders (see Supplementary Table 1) assessed the interactions between the selected goals, taking advantage of the breadth of country expertize. The participants were purposively selected based on predefined criteria of having either expertize in child health in Cambodia, or being from a non-health sector (for example water and sanitation, agriculture, infrastructure etc.) reflecting the social, economic, political, environmental, and cultural determinants of health and working in a capacity that includes multisectoral collaboration in the country.

Based on the SDG Synergies approach, groups of 5–6 people discussed direct interactions between pairs of goals, by answering a guiding question: "In the Cambodia context, if progress is made on Goal X, how does this influence progress on Goal Y?". The group arrived at a score according to the Weimer-Jehle seven-point scale[34], which ranges from strongly restricting (−3) to strongly promoting (+3). The participants also recorded a 1–2 sentence motivation for the score. The exercise was held in Khmer, official published Khmer CSDG descriptions of goals and targets were used and all documents were translated and back-translated for validity. As a basis for scoring, the participants used their expert and working knowledge, as well as a fact sheet for each goal with descriptions of the associated targets and key statistics derived from the latest Cambodia Sustainable Development Report[35]. It was emphasized that the participants should think about child health in a broad perspective, in line with the definition in Table 1, and not only on child mortality. After the first scoring of interactions, the groups double-checked their own scoring and also verified a set of interactions originally scored by another group. All identified discrepancies and differences were discussed in plenary session, where final scores were arrived at in consensus.

**Cross-impact matrix and network analysis**. All scores were directly entered into a tailor-made digital software[36] developed by the Stockholm Environment Institute, which also includes the statistical analysis features outlined below. From the final scoring of all interactions, a cross-impact matrix was developed, which served to illustrate the results and was the basis for applying network analysis. By utilizing a cross-impact matrix and keeping the analysis at the goal level, a whole of 2030 Agenda approach to child health and SDG interactions in Cambodia could be achieved. While the data presented in the cross-impact matrix provides information on the frequency of different types of interactions and how different goals influence the overall agenda, network analysis methods can be used to assert more systemic properties of the interactions. By using network analysis, where a goal is considered a node (N) and the interaction is considered a link (L) and the subsequent network can be described as $G = (N, L)$, the network can be visualized, clusters of more related goals highlighted, and the impact of certain goals and/or interactions more clearly assessed[37]. Moving beyond the direct interactions that are evident from the cross-impact matrix, analysis of the second-order interactions shows the net influence of a certain goal on the network as a whole as well as on other individual goals. Following the method described by Weitz et al. [14], the net influence (I) of a goal (g) on the network as a whole including the second-order interactions was calculated according to [Eq. 1]

$$I_g^{Total} = I_g^{1st} + \sum I^{2nd} = D_g^{Out} + \sum_{j \neq g} I_{gj} D_j^{Out} \tag{1}$$

where $I_g^{1st}$ is the influence of goal g on its closest neighbors, $I^{2nd}$ is the influence from g's neighbor's on their neighbors, $D_g^{Out}$ is the out-degree of goal g, $I_{g,j}$ is the strength of link from goal g to goal j, and $D_j^{Out}$ is the out-degree of goal j. Similarly, the aggregated second-order influence of a goal A on another goal D is estimated by

$$I_{A \to D}^{2nd} = \sum_i w_{Ai} w_{iD}$$

where I runs over all goals connecting A and D, and $w_{ij}$ is the weight on the link between goal I and goal j. A more detailed explanation of the concepts outlined above is available in the Supplementary Methods.

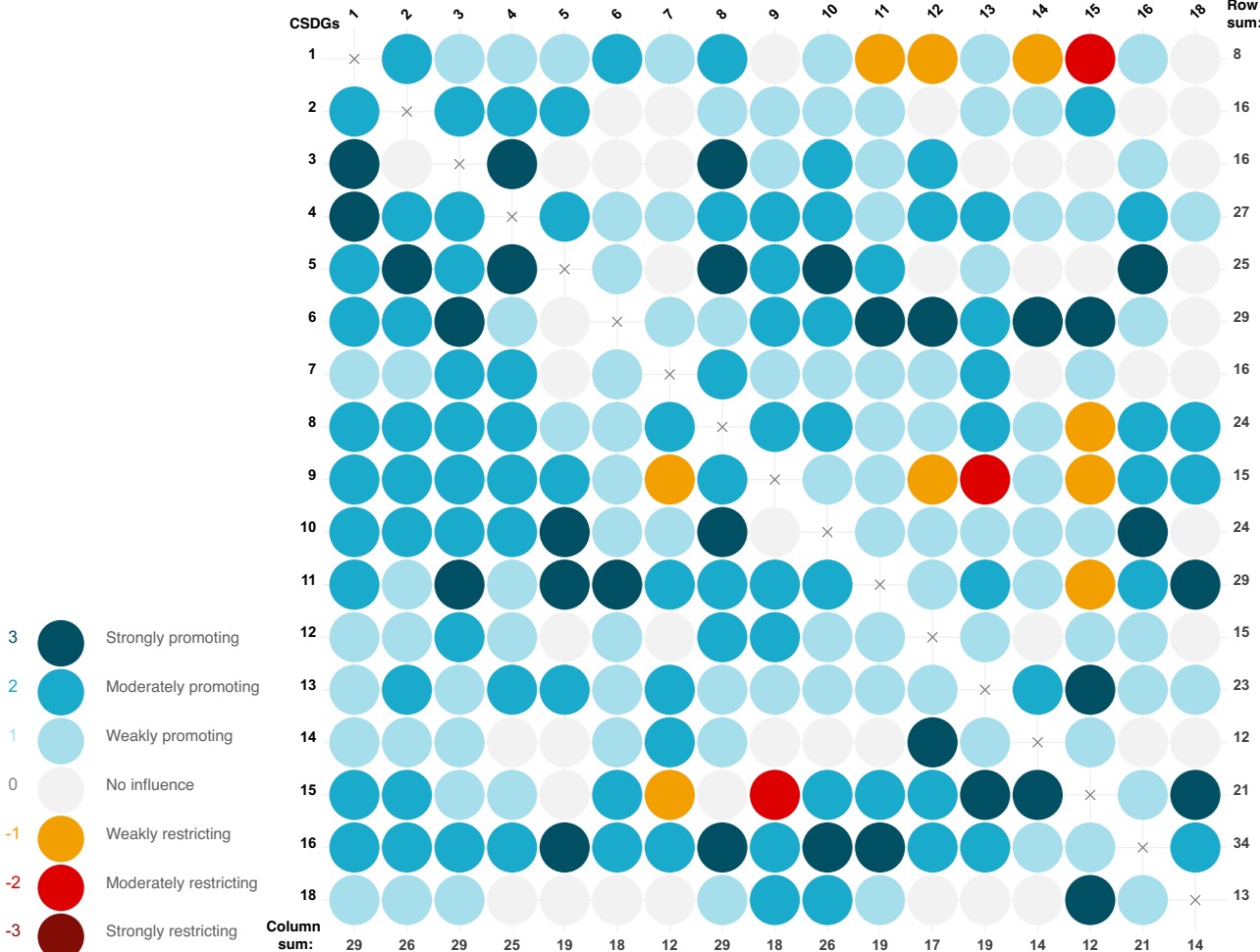

**Fig. 1 Cross-impact matrix of the 17 Cambodian Sustainable Development Goals.** Color according to scale. The row sum implies the net influence of the goal on the network, and the column net sum show the how much the goal is influenced by all other goals in the network. Cambodia Sustainable Development Goals 1 no poverty, 2 zero hunger, 3 child health, 4 quality education, 5 gender equality, 6 clean water and sanitation, 7 affordable and clean energy, 8 decent work and economic growth, 9 industry, innovation and infrastructure, 10 reduced inequalities, 11 sustainable cities and communities, 12 responsible consumption and production, 13 climate change, 14 life below water, 15 life on land, 16 peace, justice and strong institutions, and 18 mine/ERW free. The underlying data can be found in Supplementary Data 2.

**Situating of results.** Situating the results from the cross-impact matrix and network analysis is relevant to ground the analysis in the country context. Due to the lack of data on the CSDGs an overview of relevant indicators for the SDGs are provided in Supplementary Fig. 1 and Supplementary Data 1, which form the basis for a Pearson paired-observational correlation analysis to assess the trends provided in Supplementary Fig. 2. Notably, included variables were re-coded to showcase progress toward the CSDGs similar to other correlation based assessments of SDG interactions[38,39]. An overview of key developmental and child health policies are further provided in the Supplementary Fig. 3 while the annual budget expenses for each ministry between 2000-2013 is also provided in Supplementary Fig. 4 and Supplementary Fig. 5. All available data on the indicators of the CSDGs and their SDG counterpart as well as the annual budget expenses has been compiled and can be found in the Supplementary Data 1.

**Ethics approval.** The study received ethical approval from the National Ethics Committee for Health Research in Cambodia (NECHR-023) and written informed consent was obtained from all participants.

**Reporting summary.** Further information on research design is available in the Nature Research Reporting Summary linked to this article.

**Results**

**Cross-impact matrix and first and second-order interactions of the SDGs in Cambodia.** The interactions between 17 CSDGs as defined in Table 1 were scored on a seven-point scale from strongly restricting (−3) to strongly promoting (+3) by an interdisciplinary stakeholder group leading to the cross-impact matrix with 272 interactions illustrated in Fig. 1. There is a high frequency of perceived positive interactions ($n = 212$, 78%) versus negative ($n = 12$, 4%) and a substantial amount deemed to have no direct influence ($n = 48$, 18%). The row sum implies the net first order influence of the goal on the network, and the column sum shows how much the goal is directly influenced by all other goals in the network. It stands clear that CSDG 16 (peace, justice

**Table 2 Rank of goals influencing the network based on first and second-order interactions.**

| First-order interactions | | | Second-order interactions | | |
|---|---|---|---|---|---|
| Rank | Goal | Net influence | Rank | Goal | Net influence |
| 1 | 16 | 34 | 1 | 16 | 729 |
| 2 | 6 | 29 | 2 | 11 | 615 |
| 3 | 11 | 29 | 3 | 5 | 588 |
| 4 | 4 | 27 | 4 | 6 | 581 |
| 5 | 5 | 25 | 5 | 4 | 555 |
| 6 | 8 | 24 | 6 | 10 | 552 |
| 7 | 10 | 24 | 7 | 8 | 497 |
| 8 | 13 | 23 | 8 | 13 | 485 |
| 9 | 15 | 21 | 9 | 15 | 438 |
| 10 | 2 | 16 | 10 | 7 | 353 |
| 11 | 3 | 16 | 11 | 3 | 349 |
| 12 | 7 | 16 | 12 | 2 | 337 |
| 13 | 9 | 15 | 13 | 9 | 337 |
| 14 | 12 | 15 | 14 | 12 | 336 |
| 15 | 18 | 13 | 15 | 18 | 281 |
| 16 | 14 | 12 | 16 | 14 | 226 |
| 17 | 1 | 8 | 17 | 1 | 213 |

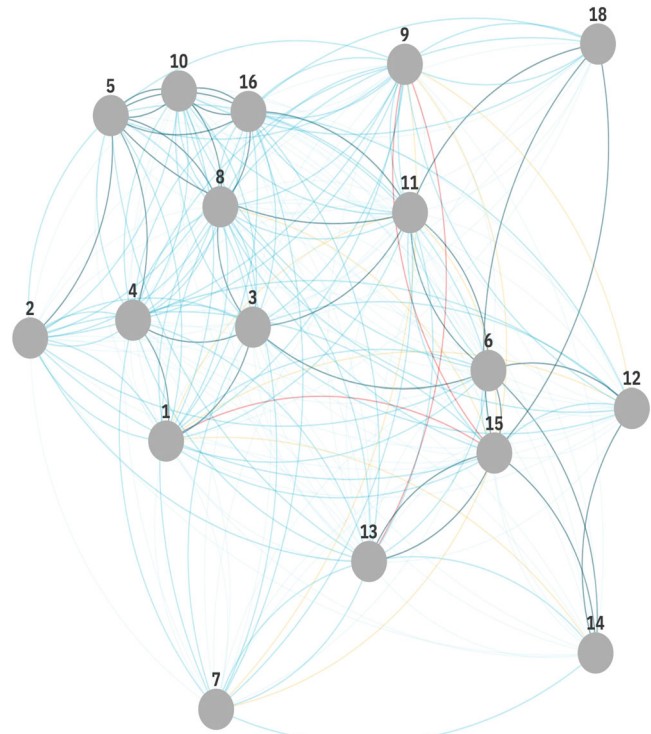

**Fig. 2 Illustration of the full network of 17 goals and 272 linkages based on the cross-impact matrix.** Cambodia Sustainable Development Goals 1 no poverty, 2 zero hunger, 3 child health, 4 quality education, 5 gender equality, 6 clean water and sanitation, 7 affordable and clean energy, 8 decent work and economic growth, 9 industry, innovation and infrastructure, 10 reduced inequalities, 11 sustainable cities and communities, 12 responsible consumption and production, 13 climate change, 14 life below water, 15 life on land, 16 peace, justice and strong institutions, and 18 mine/ERW free.

and strong institutions) has the largest first order positive influence on the network, with CSDG 11 (sustainable cities and communities) and CSDG 6 (clean water and sanitation) having the second largest direct positive influence. CSDG 1 (no poverty) has the least positive influence on the network, with negative impacts on CSDG 11 (sustainable cities and communities), 12 (responsible consumption and production), 14 (life below water) and 15 (life on land). Conversely, CSDG 1 (no poverty) together with CSDG 8 (decent work and economic growth) and CSDG 3 (child health) is promoted the most by progress on other goals, whereas CSDG 15 (life on land) is promoted the least by progress on other CSDGs. Importantly, neither the row or column sum details whether the perceived influence results from strong influence by a few targets or weak influence by many, or the distribution between positive and negative interactions.

Expanding the network from only direct first order interactions to second-order interactions, the ranks of the row sums of the goals change as illustrated in Table 2. CSDG 16 is even more clearly perceived as the most positively influencing goal of the network, while CSDG 6 (clean water and sanitation) falls from 2nd to 4th rank and CSDG 5 (gender equality) jumps from 5th to 3rd. A similar movement is made by CSDG 7 (affordable and clean energy) from 12th to 10th, while notably the bottom five goals and child health remain in their rank. The net influence in absolute terms between the ranks is however close.

The goals and their interactions can be visualized as a network, seen in Fig. 2. Although no clear clusters can be identified, it is yet again emphasized that the goals are closely interlinked but that some goals such as CSDG 7 (affordable and clean energy), 14 (life below water) and 18 (mine/ERW free) are relatively more distant from other goals in the network.

**Child health within the network**. The CSDGs in general were perceived to have a positive influence on child health in Cambodia and child health directly and positively influences many of the other CSDGs. Specifically, progress on child health was assessed to strongly promote the achievement of CSDG 1 (no poverty), 4 (quality education) and 8 (decent work and economic growth), moderately promote CSDG 10 (reduced inequalities) and 12 (responsible consumption and production) and weakly promote progress toward CSDG 9 (industry, innovation and infrastructure), 11 (sustainable cities and communities) and 16 (peace, justice and strong institutions) (Fig. 3a). The participants assessed that progress on child health does not have any direct influence on the achievement of CSDG 2 (zero hunger), 5 (gender equality), 6

(clean water and sanitation), 7 (affordable and clean energy), 13 (climate change), 14 (life below water), 15 (life on land) and 18 (mine/ERW free) in Cambodia. They acknowledged, however, that there are many second-order influences from child health on the aforementioned goals. On the other hand, child health is deemed to be strongly influenced by CSDG 6 (clean water and sanitation) and 11 (sustainable cities and communities), moderately influenced by CSDG 2 (zero hunger), 4 (quality education), 5 (gender equality), 7 (affordable and clean energy), 8 (decent work and economic growth), 9 (industry, innovation and infrastructure), 10 (reduced inequalities), 12 (responsible consumption and production), 16 (peace, justice and strong institutions) and lastly weakly influenced by CSDG 1 (no poverty), 13 (climate change), 14 (life below water), 15 (life on land) and 18 (Mine/ERW free) in Cambodia (Fig. 3b). Importantly, there does not seem to be any directly restricting interactions.

The aggregated second-order interactions found in Fig. 3c and d provide some additional insights. First, all goals, in particular CSDG 16 (peace, justice and strong institutions), have a net positive influence on child health when second-order interactions are considered. Secondly, there seems to be a potentially important positive feedback-loop, whereby improving child health in itself lead to the promotion of child health through promoting interactions of other CSDGs. Thirdly, although the second-order interactions are generally positive, they show a negative influence of child health on CSDG 15 (life on land) not showcased before. This implies an important trade-off that must be handled, which could have been overlooked if researchers and policy makers only focus on direct interactions.

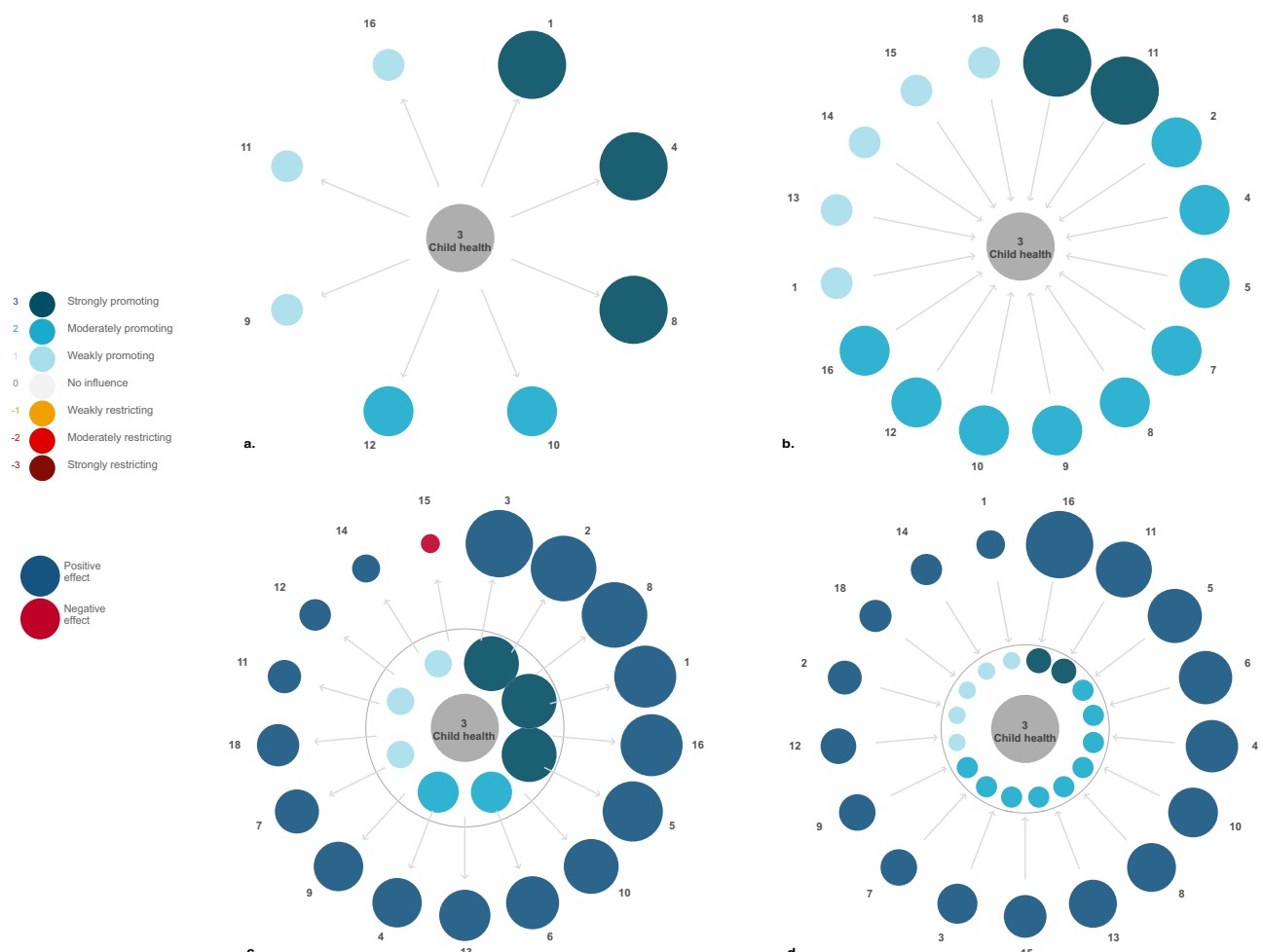

**Fig. 3 The Cambodia Sustainable Development Goals from the perspective of child health. a**, **b**: (**a**) Illustrate the first-order influence of child health on the CSDGs and (**b**) vice versa. **c**, **d**: (**c**) Illustrate the second-order influence of child health on the CSDGs and (**d**) vice versa. Color according to scale. Cambodia Sustainable Development Goals 1 no poverty, 2 zero hunger, 3 child health, 4 quality education, 5 gender equality, 6 clean water and sanitation, 7 affordable and clean energy, 8 decent work and economic growth, 9 industry, innovation and infrastructure, 10 reduced inequalities, 11 sustainable cities and communities, 12 responsible consumption and production, 13 climate change, 14 life below water, 15 life on land, 16 peace, justice and strong institutions, and 18 mine/ERW free.

## Discussion

This study constitutes the first attempt at an empirical investigation at a national level into the interactions between child health and the SDGs. In general, the interactions were perceived to be synergistic between the child health and the CSDGs, and progress on CSDG 16 (peace, justice and strong institutions) could have the largest potential to contribute to the achievement of the CSDGs, both when it comes to first and second-order interactions in Cambodia. All goals were deemed to positively influence child health in some way and child health was thought to have a promoting influence on the achievement of other goals except for CSDG 15 (life on land). These findings are in line with similar assessment noting the overall positive influence of good health and well-being on the possibility of achieving other sustainable development outcomes[12,17,20].

The SDGs and the locally adapted CSDGs offer an overarching framework that encompasses many of the determinants of child health. Within the Cambodian context, our analysis suggests that in line with the literature on the advancements in child mortality, stakeholders perceive that child health is heavily dependent on progress in other sectors. Further, comparing with data on the key indicators that exist (Table 3) showcase that progress on child health has been positively correlated with a number of CSDGs in

Cambodia, but that progress has not coincided with progress toward CSDG 12, 13, 14, 15, and 16 (See Supplementary Fig. 2). Interestingly, there were no restricting interactions found at the first or second-order analysis on child health from making progress on any of the other goals in our analysis, which might suggest the fact that child health in general is closely related to the social determinants of health which the other goals reflect. When considering second-order interactions, CSDG 16 (peace, justice and strong institutions) was perceived to have the largest net positive influence on child health. An example of how effective institutions in Cambodia positively influence child health is the success of the community based poverty identification system ID-Poor which has served as a platform for multisectoral collaboration on various health and non-health issues targeting the poor in Cambodia and strengthening institutional frameworks[29]. However, a continuous high rate of out of pocket spending have impeded the advancement and coverage of health services[40] and ID-Poor beneficiaries who are disabled do not yet have full access to health care services and other social protection schemes[41]. During the same time the roles and engagement of civil society in health service delivery and social support have decreased[42–44]. Expanding social protection systems and strengthening local institutions together with increased collaboration with civil society might help to accelerate gains in child

**Table 3 Overview of key Cambodian sustainable development indicators.**

| CSDG | Indicator description | 2000 | 2005 | 2010 | 2015 | 2019 | Source |
|---|---|---|---|---|---|---|---|
| 1 | Proportion of population living below the national poverty line (%) | 50 (2003) | 45 (2006) | 22.1 | 14 (2014) | | SDG Indicator database |
| 2 | Prevalence of undernourishment (%) | 24 (2001) | 17 | 13 | 8.9 | 6.2 | SDG Indicator database |
| 3 – Child health | Under-five mortality rate, by sex (deaths per 1000 live births) | 106 | 65 | 44 | 32 | 27 | SDG Indicator database |
| | Proportion of children under 5 years moderately or severely stunted (%) | 49 | 43 | 39.8 | 32 (2014) | | SDG Indicator database |
| | Proportion of children under 5 years moderately or severely wasted (%) | 17 | 8,5 | 11 | 9,7 (2014) | | SDG Indicator database |
| 4 | Completion rate at primary level (%) | 34 | 59 | 71 | 72 (2014) | | SDG Indicator database |
| | Completion rate at secondary level (%) | 17 | 27 | 37 | 41 (2014) | | SDG Indicator database |
| 5 | Proportion of seats held by women in legislation institutions (%) | 8.2 | 9.8 | 21 | 20 | 20 | SDG Indicator database |
| 6 | Proportion of population using safely managed drinking water services (%) | 17 | 19 | 22 | 25 | 27 | SDG Indicator database |
| | Proportion of population with basic handwashing facilities on premises (%) | | | 63 | 67 | 73 | SDG Indicator database |
| 7 | Proportion of population with access to electricity, by urban/rural (%) | 17 | 21 | 31 | 69 | 93 | SDG Indicator database |
| 8 | GDP per capita (current US$) | 301 | 474 | 785 | 1162 | 1643 | World Bank |
| 9 | Proportion of population covered by at least a 3 G mobile network (%) | 17 | | 43 (2009) | 70 | 85 | SDG Indicator database |
| 10 | Gini index disposable income (0–100) | 37 | 37 | 36 | | | Standardized World Income Inequality Database |
| 11 | Proportion of urban population living in slums (%) | | 79 | | 55 (2014) | 45 (2018) | SDG Indicator database |
| 12 | Domestic material consumption per capita (tonnes) | 2.2 | 2.7 | 6.1 | 4.9 | 5.3 (2017) | SDG Indicator database |
| 13 | Domestic fossil fuel consumption per capita (tonnes) | 0.1 | 0.1 | 0.1 | 0.2 | 0.2 (2017) | SDG Indicator database |
| 14 | Sustainable fisheries as a proportion of GDP (%) | | | 1,1 (2011) | 0,8 | 0,6 (2017) | SDG Indicator database |
| 15 | Forest area as a proportion of total land area (%) | 61 | | 60 | 50 | 46 | SDG Indicator database |
| 16 | Voice and Accountability (ranges from approximately −2.5 (weak) to 2.5 (strong) governance performance) | −0.8 | −1.0 | −0.9 | −1.1 | −1.2 | The Worldwide Governance Indicators |

health and well-being even further. Conversely, progress on child health is deemed to be strongly promoting progress directly on a few key CSDGs, including CSDG 1 (no poverty), CSDG 4 (quality education) and CSDG 8 (decent work and economic growth). The relationships between child health and these policy areas have been characterized as positive and important in multiple studies[4,45,46] and is in line with historic trends seen in Table 3, as such our results add empirical country level evidence to the knowledge base. When deciphering second-order interactions, our findings show a positive feedback-loop regarding child health in Cambodia, where progress on child health and well-being in itself through the promoting effect on other goals, leads to further progress in child health. Further, child health has a net promoting second-order influence on all other goals except CSDG 15 (life on land) on which it has a relatively small negative net influence. This is derived from the fact that child health is perceived to promote CSDGs that in total have a net restricting influence on CSDG 15 (life on land), primarily through CSDG 1 (no poverty), 8 (decent work and economic growth), 9 (industry, innovation and infrastructure) and 11 (sustainable cities and communities) in our analysis. These interactions might be explained by the apparent trade-offs between land conservation efforts and progress on other development goals such as reducing poverty and increasing agricultural productivity in Cambodia[47,48]. When examining the trends in the historic data provided in Table 3 combined with (i) the targets set for 2030 of restoring forests to around 50% of the total land (CSDG 15.1.1) and (ii) simultaneously keeping the 7% annual growth rate of real GDP per capita (CSDG 8.1.1) and (iii) almost eradicating extreme poverty[32] it becomes evident that there might be some cause for this concern by the participants especially given the decreasing annual spending of the Ministry of Agriculture, Forestry and Fishery (Supplementary Fig. 5). Importantly, stakeholders considered child health to not have any direct influence on the possibility to make progress on CSDG 15 (life on land), and it is clear that the two goals could have synergistic potential[49]. The indivisible and complex relationships between sustainable development outcomes showcase that trade-offs that are not apparent at first glance might have implications for the overall achievement of the agenda.

When applying novel methods for answering research questions adequate reflections on the merits of the analysis are due. The SDG Synergies approach hinges crucially on the selection of goals or targets to analyze, the group of participants that are tasked to make the scoring and the quality of the scoring process. Moving from the goal level to the target level within the SDG framework would probably alter the results. Further, the goals are broadly defined and can be interpreted in different ways when making assessments. A different set of in-country experts, and including private sector representatives, might therefore have judged the interactions in another way. Nevertheless, by clearly framing the goals and utilizing local stakeholders' expert judgment through a double-scoring process leading up to a consensus choice ensured that relevant and relatively unbiased scores were identified. The scoring of the interactions, however, does not rely on any pure quantitative assessment and should therefore be interpreted with caution. While grounding the interactions found in the country context and in the available data allows for a deeper understanding of the relationships found, it is not possible to derive a definitive causal direction for each individual interaction or the network as a whole. The primary results from this study, which are focused on systemic patterns from the perceptions of stakeholders, are only a small contribution to the knowledge base and would benefit from being complemented with research focusing on more specific goals, zooming in on a smaller regional or district geographical area and perhaps include a more formal assessment of how the interactions noted by the stakeholders corresponds to the policies formed and implemented

to achieve better child health in Cambodia. Crucially, the advantages of contextualization and applicability must be weighed against the desired generalizability when using the SDG Synergies approach, as findings become harder to generalize across political, economic, geographical, and social settings[14,20]. Altogether, the strengths and limitations of the method of the results reflect the complexity of the 2030 Agenda itself.

An integrated analysis that transcend sectoral boundaries is necessary to form a bridge between science and policy making for sustainable development in general[50] and child health specifically[20,51]. As our findings illustrate, progress on several CSDGs are important for child health and well-being, while child health in itself promotes progress for sustainable development in Cambodia. In particular, policy makers should consider direct and indirect interactions between child health and CSDG 16 (peace, justice and strong institutions) given the strong net positive effect on child health and the worryingly negative trend over the last decade, while being observant of the net negative effect of progress on child health on CSDG 15 (life on land).

Beside the findings in themselves, the participatory approach of the SDG Synergies approach was greatly appreciated by the included stakeholders and served as an opportunity to meet and discuss multisectoral issues and potential partnerships, framing the discussions around sustainable development, synergies and trade-offs in a common language. Framing key common determinants and prioritizing multisectoral efforts are vital to ending preventable child deaths[9]. With the risk of competing priorities and limited funding to reach the SDGs, continuous assessment and dialogue of potential synergies and trade-offs are essential to overcome bottlenecks and promote policy relevance. The SDG Synergies approach can serve as one tool for better governance on these issues, allowing also for comparison of interactions found with actual policy priorities and investments[52]. Overall, a participatory approach such as the SDG Synergies which allows for a systematic assessment of the interactions surrounding the SDGs and child health can provide novel insights on how complex relationships play out on a country level. With the need to further place the child in the centre of the SDGs[51] and given the multifaceted challenges facing global child health[49] this understanding will be vital for informing policy coherence and exploring innovative multisectoral partnerships that can accelerate the work toward achieving the 2030 Agenda in general and the betterment of global child health in particular.

## Data availability
Source data are included in this published article in Supplementary Data 1 and Supplementary Data 2.

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

## Acknowledgements

The authors would like to express our thankfulness to all the participants that took part in the study. We are further grateful for the assistance from Mr. Tim Vora, Mr. Sou Veasna, Mr. Bunhoeuth Thou in helping to lead the workshop, Mr. Roy Nijhof for unwavering technical support and Mr. Mark Debackere for graciously lending time and effort making this study possible despite the COVID-19 pandemic.

## Author contributions

D.H. was responsible for study design, data analysis, interpretation of data and for the writing process including writing a first draft of the paper and approval of the final draft. T.C. and S.S. contributed to data collection and revised the paper and approved the final draft. L.J. contributed to study design, data analysis, revised the paper and approved the final draft. H.N. and G.T. contributed to interpretation of data, revised the paper and approved the final draft. M.N. contributed to study design, paper revision and approved the final draft. T.A. contributed to the study design, data interpretation, paper revision and approved the final draft.

## Funding

## Competing interests

The authors declare no competing interests.

**Additional information**

