## [Peer Review File · Communications Medicine]

Reviewers' comments:

Reviewer #1 (Remarks to the Author):

This manuscript reviews the status of various SDGs in Cambodia through a network analysis and one of SDG linkages. The assessment is mainly based on a qualitative assessment and ranking by a stakeholders group and while linkages and relative importance was assessed, no information is provided on any validation or supportive strategies.

It could have helped to see coverage and co-coverage data for various goals and key indicators to validate some of the assumptions of tracking and change. The basic premise of how various SDGs relate to each other is well known and what this country-level analysis could have added is evidence on performance, investments and gaps. In the absence of that information, I find the network analysis complex and duplicative of past work. The paper could have dealt with the important issue of policy relevance for implementation and addressing bottlenecks but I don't quite see that percolating through.

Reviewer #2 (Remarks to the Author):

The authors analyze a quite interesting topic, that of the assessment of how SDGs influence a desired outcome, in this case, child health in Cambodia. The paper is well written and I suggest acceptance with minor revision.

- Explain the limitations of the SDG Synergies methodology
- Child health here refers to children under 18 years old, although the introduction focuses mainly on under-five mortality. The authors need to elaborate on how the respondents perceived "child health" in the survey
- How the authors conclude that child health promotion and SDG promotion can both be the cause and effect of each other needs to be further elaborated. It is unclear how this conclusion is derived from the applied method
- Even though the limitations of the study are discussed, it is unclear which SDGs should be a policy priority for child health promotion.

Reviewer #3 (Remarks to the Author):

Brief summary of the manuscript

The work describes a consultation session with Cambodian officials and experts to delineate, from a localized perspective, the relationships between different SDG (or targets of SDGs). The authors then use the outputs of that consultation to develop a network model to quantify the first and second order interactions between each of them and thereby demonstrate a likely synergy in the SDG agenda.

Overall impression of the work

The paper is well written and this is an interesting, novel, and useful heuristic for how to think about the SDG agenda. Based on the length of the piece, I was expecting a couple more layers to the analysis (e.g. evaluating the framework, comparing against recent policy and/or expenditure in Cambodia). It is still a valuable contribution even without those additional components, but in that case I believe the concept of this paper could be summarized in a much shorter piece, which I think

would actually allow it to have greater impact due to more rapid digestibility.

Specific comments, with recommendations for addressing each comment

1. Minor comment: Line 83 and 146 – “... a myriad of ...” is grammatically incorrect. Should be simply “... myriad ...” and it is plural = “are myriad...”
2. It took me until the very end of the results section to realize that this is a study based entirely on quantification of expert opinion. Recommend to make this a bit more clear in the abstract and introduction.
3. While reading, I was expecting an additional section of the analysis to actually apply this framework to the levels and trends of different SDG indicators in Cambodia to evaluate if there is quantitative face validity to the proposed framework. Is there evidence in the recent past trends in Cambodia to support the synergies suggested in the authors’ framework?
4. Another thing that is missing in the discussion is a clear recommendation from the authors about how to adopt this assessment into the policymaking arena. The relevance of adoption, and value proposition for repeating this exercise in other countries (or subnational policymaking units), could be to compare the synergies suggested by the framework with recent policy efforts and/ or governmental spending patterns in recent decades.
5. If neither 3 nor 4 is possible or feasible, this would for me reinforce the notion that this paper would be of more impact if substantially pared down to very succinctly describe the synergistic framework.

Reviewer comments

Reviewer Comments	Authors' Response and Revisions
Reviewer 1	
General comment(s)	
This manuscript reviews the status of various SDGs in Cambodia through a network analysis and one of SDG linkages. The assessment is mainly based on a qualitative assessment and ranking by a stakeholders group and while linkages and relative importance was assessed, no information is provided on any validation or supportive strategies.	Dear Reviewer, Thank you for your insightful comments. We have built upon your feedback and added substantial amount of secondary data and analysis. Please find our detailed responses below. Please note that the page numbers and line numbers in the responses are referring to the manuscript without track-changes.
Specific comments	
It could have helped to see coverage and co-coverage data for various goals and key indicators to validate some of the assumptions of tracking and change.	We have collected the data that exist on the SDGs and Cambodian SDGs, unfortunately there are significant gaps which makes it hard to assess the interactions between the Cambodian SDGs in a quantitative fashion (time series analysis/regression or similar). However, to complement the results of the study we have added an overview of key indicators in Table 3, to form an overall sense of the trends and coverage of various CSDGs over time. Additionally, we provide the following in the Supplement Information:  i) Overview of child mortality trends by age group in Cambodia. ii) Correlation analysis to showcase how the CSDGs have historically compared, using a similar approach that others who have tried to find synergies and trade-offs between the SDGs have used (Pradhan, P. Nat. Sustain. 2, 171–172 (2019) & Kroll, C., Warchold, A. & Pradhan, P. Palgrave Commun. 5, 1–11 (2019)) iii) Overview of the policy development (general and child health specific). iv) Outline of the government annual expenses by line ministries between 2000-2013 (longest period with comparable data available). Lastly, we have compiled all available data for Cambodian SDG indicators as well as the targets for 2020, 2025 and 2030 set by the government of Cambodia in 2018 (the most updated official

	document), which gives an indication of the focus and ambitions of the Cambodian SDGs and their targets and made it publicly available (link: https://ki.se/en/gph/research-projects, under headline “Multi-sectoral and policy research”). We have used this secondary data sources to try and compare the key findings of the study with the available data throughout the manuscript.
The basic premise of how various SDGs relate to each other is well known and what this country-level analysis could have added is evidence on performance, investments and gaps. In the absence of that information, I find the network analysis complex and duplicative of past work.	Indeed, individual interactions between various goals have been assessed in various ways before, however we believe the study is novel in two distinct ways. First, we provide a systematic approach to the interactions between the Cambodian SDGs and child health, not conducted before and can conclude that there is a high frequency of perceived positive interactions (n=212, 78%) versus negative (n=12, 4%) and a substantial amount deemed to have no direct influence (n=48, 18%). Generic interactions have been previously assessed, but since the interactions are highly context specific we deem our analysis an important and relevant contribution to the evidence base. Second, through the use of network theory we can showcase the second-order effects, capturing the impact of each Cambodian SDG on child health and vice versa in a much fuller way. The drawback or limitation of this approach is the added complexity, however we still believe that the use of second order interactions enable policy makers to see beyond individual goals or interactions, and get a broader sense of how the Cambodian SDGs are interlinked which allows for new partnerships and policy approaches. To complement the results from the study, we have included the key policy developments in Cambodia, as well as compiled the available annual expenses of the government by line ministry, which we provide in the Supplemental Information and have incorporated into the discussion on the key interactions. This, together with the correlation analysis and data provided on the indicators, gives an indication of the performance, investments and current gaps for reaching the Cambodian SDGs.

	The above act to give further context on the results from the study, and to allow us (and the reader) to critically assess the relationships found and conclusions drawn from the study. To the best of our knowledge, there have been no similar article to provide this comprehensive data and analysis of Cambodian SDG interactions with a focus on child health in Cambodia.
The paper could have dealt with the important issue of policy relevance for implementation and addressing bottlenecks but I don't quite see that percolating through.	This is a very good point, the identified trade-offs can be considered as bottlenecks to implementation, while the participatory approach of the SDG Synergies method that engage policy makers and country experts directly in these issues allows for a dialogue and common policy language when discussing and thinking about the Cambodian SDGs and implementing policies. We have added this more clearly in i) the introduction on page 4, line 86-90: “To some extent, the SDG Synergies approach integrates real world human behaviour into prioritisation and decision making models which is necessary for understanding complex context dependent systems^{24,25}, forming a bridge across sectors and promoting evidence informed policy, particularly given the absence of quality quantitative data to assess the SDGs²⁶. “ ii) Discussion on page 11, line 265-268 “Beside the findings in themselves, the participatory approach of the SDG Synergies approach was greatly appreciated by the included stakeholders and served as an opportunity to meet and discuss multisectoral issues and potential partnerships, framing the discussions around sustainable development, synergies and trade-offs in a common language.” iii) Discussion on page 11, line 269-272 “With the risk of competing priorities and limited funding to reach the SDGs, continuous assessment and dialogue of potential synergies and trade-offs are essential to overcome bottlenecks and promote policy relevance.”

Reviewer 2	
General comment(s)	
The authors analyze a quite interesting topic, that of the assessment of how SDGs influence a desired outcome, in this case, child health in Cambodia. The paper is well written and I suggest acceptance with minor revision.	Dear Reviewer, Thank you for the comments, based on very important points brought up by the reviewers, we have made substantial changes that we think have helped to improve the quality and scientific value of the manuscript. Please note that the page numbers and line numbers in the responses are referring to the manuscript without track-changes.
Specific comments	
Explain the limitations of the SDG Synergies methodology	In short, three main limitations of the SDG Synergies method are:  i) Limitation of the number of interactions that the participants/stakeholders can feasibly assess ii) The subjectivity of the scoring process iii) The need for grounding the assessment of interactions in context hampers generalisability of the results We have added and clarified this in the discussion section on page 10, lines 234-245. “The SDG Synergies approach hinges crucially on the selection of goals or targets to analyse, the group of participants that are tasked to make the scoring and the quality of the scoring process. Moving from the goal level to the target level within the SDG framework would probably alter the results. Further, the goals are broadly defined and can be interpreted in different ways when making assessments. A different set of in-country experts, and including private sector representatives, might therefore have judged the interactions in another way. Nevertheless, by clearly framing the goals and utilising local stakeholders’ expert judgment through a double-scoring process leading up to a consensus choice ensured that relevant and relatively unbiased scores were identified. The scoring of the interactions, however, does not rely on any pure quantitative assessment and should therefore be interpreted with caution. While grounding the interactions found in the country context and in the available data allows for a deeper understanding of the relationships found, it is not possible to derive a definitive causal direction for

	each individual interaction or the network as a whole.”
Child health here refers to children under 18 years old, although the introduction focuses mainly on under-five mortality. The authors need to elaborate on how the respondents perceived "child health" in the survey	Thank you for this important comment. In the introduction we have highlighted under-five mortality since this is one of the indicators with most data available. In order to try and keep the introduction length relatively short, we have added some emphasis on the multidimensional poverty of children in Cambodia. Page 5, lines 99-102: “However, multidimensional poverty and non-monetary deprivation such as overcrowded housing, suboptimal water and sanitation facilities and lack of school attendance are still prevalent with almost half of all children under 18 years of age experienced three or more deprivations in 2018³¹.” The participants were given the definition “In line with the WHO definition of health and the United Nations Convention on the Rights of the Child, child health is defined as a state of complete physical, mental and social well-being and not merely the absence of disease or infirmity among human beings below 18 years” (See Table 1), hence quite a broad perception. The participants were instructed to think about child health through this definition, and not only on mortality. We have added this aspect on page 14, lines 325-327: “it was emphasised that the participants should think about child health in a broad perspective, in line with the definition in Table 1, and not only on child mortality.”
How the authors conclude that child health promotion and SDG promotion can both be the cause and effect of each other needs to be further elaborated. It is unclear how this conclusion is derived from the applied method	Thank you for the observation, we want to clarify that we do not try to directly assess or derive causality through the use of the SDG Synergies method. Unfortunately, to date, there is not enough data to perform traditional quantitative time series/regression analysis, which could provide some more guidance on the causal aspects of the relationships found in our study. We try to amend this through conducting correlation analysis and providing all data available in Supplemental Information (see responses to other reviewers).

	However, by including the second order interactions we move one step further from the direct interaction between CSDGs and child health, and analyse the next step, or the second “ripple” in the network. This allows us to illustrate how progress on one Cambodian SDG/child health affects the ability to make progress on all other Cambodian SDGs/child health and the original Cambodian SDG/child health. We have clarified this in the limitation section in the discussion, page 10-11 lines 243-250. “While grounding the interactions found in the country context and in the available data allows for a deeper understanding of the relationships found, it is not possible to derive a definitive causal direction for each individual interaction or the network as a whole. The primary results from this study, which are focused on systemic patterns from the perceptions of stakeholders, are only a small contribution to the knowledge base and would benefit from being complemented with research focusing on more specific goals, zooming in on a smaller regional or district geographical area and perhaps include a more formal assessment of how the interactions noted by the stakeholders corresponds to the policies formed and implemented to achieve better child health in Cambodia.” We have also re-phrased the conclusion statement to exclude the notion of causation on page 11 lines 258-259. “As our findings illustrate, progress on several CSDGs are important for child health and well-being, while child health in itself promotes progress for sustainable development in Cambodia.”
Even though the limitations of the study are discussed, it is unclear which SDGs should be a policy priority for child health promotion.	Given that Cambodian SDG 16 (peace, justice and strong institutions) have the largest net positive effect on child health, and the negative trend of this Cambodian SDG over the last decade, we have highlighted this Cambodian SDG in the last section. However trade-offs might be equally important, so policy makers need to observe the effects of child health improvements on Cambodian SDG 15 (life on land). This has been clarified on page 11, lines 259-263:

	“In particular, policy makers should consider direct and indirect interactions between child health and CSDG 16 (peace, justice and strong institutions) given the strong net positive effect on child health and the worryingly negative trend over the last decade, while being observant of the net negative effect of progress on child health on CSDG 15 (life on land).”
Reviewer 3	
General comment(s)	
The work describes a consultation session with Cambodian officials and experts to delineate, from a localized perspective, the relationships between different SDG (or targets of SDGs). The authors then use the outputs of that consultation to develop a network model to quantify the first and second order interactions between each of them and thereby demonstrate a likely synergy in the SDG agenda. Overall impression of the work The paper is well written and this is an interesting, novel, and useful heuristic for how to think about the SDG agenda. Based on the length of the piece, I was expecting a couple more layers to the analysis (e.g. evaluating the framework, comparing against recent policy and/or expenditure in Cambodia). It is still a valuable contribution even without those additional components, but in that case I believe the concept of this paper could be summarized in a much shorter piece, which I think would actually allow it to have greater impact due to more rapid digestibility.	Dear Reviewer, We appreciate your comments, to which we have provided detailed responses below. Please note that the page numbers and line numbers in the responses are referring to the manuscript without track-changes.
Specific comments	
1. Minor comment: Line 83 and 146 – “... a myriad of ...” is grammatically incorrect. Should be simply “... myriad ...” and it is plural = “are myriad...”	Well observed, this has been corrected.
2. It took me until the very end of the results section to realize that this is a study based entirely on quantification of expert opinion. Recommend to make this a bit more clear in the abstract and introduction.	We have tried to clarify this in the abstract and in the introduction. Please see page 4, lines 83-90: “The method quantifies expert opinions through the scoring of the interactions, and although the subjectivity of the SDG Synergies approach can be in contrast to the classical paradigm of rational and data driven decision making^{21,22}, real world prioritization processes are influenced by myriad of factors and

	biases^{16,23,24}. To some extent, the SDG Synergies approach integrates real world human behaviour into prioritisation and decision making models which is necessary for understanding complex context dependent systems^{24,25}, forming a bridge across sectors and promoting evidence informed policy, particularly given the absence of quality quantitative data to assess the SDGs²⁶. ”
3. While reading, I was expecting an additional section of the analysis to actually apply this framework to the levels and trends of different SDG indicators in Cambodia to evaluate if there is quantitative face validity to the proposed framework. Is there evidence in the recent past trends in Cambodia to support the synergies suggested in the authors’ framework?	A very good point, also raised by the first reviewer. Please see response to the comments above which outlines the added data and analysis. Overall, the historical trends of the Cambodian SDGs provide additional explanation value to the interactions, but do not at a first glance support all scores of the interactions. We have included comparisons in the manuscript for the key interactions in the discussion section. Page 9, lines 199-207: “ Indeed, when examining historical trends Cambodia has not made progress in this area (Table 3), and perhaps most remarkably the three CSDG indicators for measuring progress towards CSDG 16 set by the government do not include indicators corruption or accountable institutions³². The gradual weakening of democratic institutions over the last decade^{33,34} including a trend of limited control of corruption and shrinking space for civil society in Cambodia³⁵⁻³⁷ risks undermining progress on child health. In our study, the emphasis on the importance of CSDG 16 (peace, justice and strong institutions) for making progress on child health by the participants reflect an opportunity lost for accelerating gains in child health and well-being.” Page 9, lines 209-212: “The relationships between child health and these policy areas have been characterized as positive and important in multiple studies^{4,38,39} and is in line with historic trends seen in Table 3, as such our results add empirical country level evidence to the knowledgebase.” Page 9-10, lines 219-227: “These interactions might be explained by the apparent trade-offs between land conservation

	efforts and progress on other development goals such as reducing poverty and increasing agricultural productivity in Cambodia ^{40,41}. When examining the trends in the historic data provided in Table 3 combined with i) the targets set for 2030 of restoring forests to around 50% of the total land (CSDG 15.1.1) and ii) simultaneously keeping the 7% annual growth rate of real GDP per capita (CSDG 8.1.1) and iii) almost eradicating extreme poverty ³² it becomes evident that there might be some cause for this concern by the participants especially given the decreasing annual spending of the Ministry of Agriculture, Forestry and Fishery (Supplement Information Table 2 and Figure 4).”
4. Another thing that is missing in the discussion is a clear recommendation from the authors about how to adopt this assessment into the policymaking arena. The relevance of adoption, and value proposition for repeating this exercise in other countries (or subnational policymaking units), could be to compare the synergies suggested by the framework with recent policy efforts and/ or governmental spending patterns in recent decades.	Indeed, the comparison with recent spending patterns and policy priorities situates the results in the context and adds significant exploratory value. We have tried to include this comparison to the best of our ability given the data available in the manuscript, while also providing a comprehensive policy overview and detailed budget expense pattern by line ministry in the Supplemental Information.
5. If neither 3 nor 4 is possible or feasible, this would for me reinforce the notion that this paper would be of more impact if substantially pared down to very succinctly describe the synergistic framework.	We have opted to try and incorporate quantitative data available as recommended by this reviewer and others for adding explanatory value to the paper. This makes the manuscript a bit longer, but we have tried to strike a balance between complexity, coherence and readability.

REVIEWERS' COMMENTS:

Reviewer #1 (Remarks to the Author):

I believe the authors have done a reasonably robust job of responding to the reviews and critique. The revised manuscript is much improved and I am happy to nod approval

Reviewer #2 (Remarks to the Author):

The authors have sufficiently addressed the raised comments and I suggest acceptance of the manuscript.

Reviewer #3 (Remarks to the Author):

The authors have done a good job of responding to comments and strengthening the manuscript.

"... myriad of..." still appears in the final PDF, but i presume that can easily be corrected.